# Recent Advances in Polymers for Potassium Ion Batteries

**DOI:** 10.3390/polym14245538

**Published:** 2022-12-17

**Authors:** Xingqun Zhu, Rai Nauman Ali, Ming Song, Yingtao Tang, Zhengwei Fan

**Affiliations:** 1School of Materials and Chemical Engineering, Xuzhou University of Technology, Xuzhou 221018, China; 2School of Materials Science and Engineering, Jiangsu University, Zhenjiang 212013, China

**Keywords:** polymers, Potassium-ion batteries, electrode materials, electrolytes, binders

## Abstract

Potassium-ion batteries (KIBs) are considered to be an effective alternative to lithium-ion batteries (LIBs) due to their abundant resources, low cost, and similar electrochemical properties of K^+^ to Li^+^, and they have a good application prospect in the field of large-scale energy storage batteries. Polymer materials play a very important role in the battery field, such as polymer electrode materials, polymer binders, and polymer electrolytes. Here in this review, we focus on the research progress of polymers in KIBs and systematically summarize the research status and achievements of polymer electrode materials, electrolytes, and binders in potassium ion batteries in recent years. Finally, based on the latest representative research of polymers in KIBs, some suggestions and prospects are put forward, which provide possible directions for future research.

## 1. Introduction 

The demand for micro and small mobile intelligent gadgets (such as smartphones, tablets, wearable technology, and other similar devices) and huge power vehicles rises as human life progressively enters a new era of intelligence, which also raises the need for energy storage systems [1]. Energy can be stored using batteries. One of these is the secondary battery (rechargeable battery), which has a high energy density, exceptional portability, a long life, and other benefits. It is also an effective and recyclable method of energy conversion and storage [2]. It serves as the primary power source for portable electronic devices and offers a significant solution to energy scarcity and environmental issues. It is a priority area for the major developed countries in the world [3]. The development of new secondary batteries with high specific energy density, power density, and long cycle life is a hot research field in the international frontier [4,5,6].

Commercially speaking, lithium-ion batteries (LIBs) are the most widely used secondary batteries that are essential to our daily lives [7,8]. However, finding alternate secondary battery systems that can replace LIBs is essential because of the problem of worldwide scarcity of lithium resources brought on by the rising demand for lithium [9]. In contrast, potassium (K), which is in the same first main family as lithium and has chemical properties similar to those of lithium, has the advantages of abundant resources, widespread distribution, and inexpensive cost [10,11]. Figure 1a displays the elemental abundance in the Earth’s crust. Potassium-ion batteries (KIBs) have a similar cell structure and operating system to LIBs (Figure 1b) [10]. Due to solvation effects in electrolytes, K also has a lower redox potential [11] and a higher diffusion coefficient than sodium [12,13]. In addition, KIBs development can benefit from the lowest potential for K^+^/K redox couple in some organic electrolytes [14]. For example, even though the standard hydrogen electrode (SHE) of K^+^/K (−2.936 V) is in between Li^+^/Li (−3.040 V) and Na^+^/Na (−2.714 V) couples, both theoretical calculations and experiments have demonstrated that the K^+^/K redox couple has a reduction potential of −2.88 V in organic solvents such as propylene carbonate (PC), lower than Li^+^/Li (−2.79 V) and Na^+^/Na (−2.56 V) [15,16]. This could lead to a wider electrochemical voltage window, resulting in a high energy density for KIBs. The K^+^ exhibits much weaker Lewis acidity, leading to smaller solvated ions as compared to Li^+^ and Na^+^ [17]. Thus, the conductivity and amounts of solvated and transported K^+^ are higher than those of Li^+^ and Na^+^. In addition, the lower desolvation energy of K^+^ could lead a faster diffusion kinetics across the electrolyte/electrode interface [18,19]. The capacity of potassium to intercalate into graphite, the cheapest and most popular anode material for rechargeable metal-ion batteries, is another significant benefit of potassium over sodium [20]. Another favorable advantage is that potassium does not form an alloy when it is in contact with aluminum at lower voltages, which can reduce battery production costs by replacing copper with an aluminum substrate on the anode side [21]. Thus, in recent years, the study, development, and use of KIBs in the energy sector have drawn a lot of attention. When compared to LIBs, KIBs are clearly more affordable, are seen as LIBs replacements, and have promising futures as large-scale energy storage batteries. 

Polymers, which include polymer electrode materials (PEMs), polymer electrolytes (PEs), polymer binders, etc., play a key role in the potassium (K) battery system as one of the important organic components. Electrode materials occupy the core in KIBs. The development of high-performance electrode materials may encourage the earliest feasible commercialization of KIBs. Polymers are still being investigated as potential KIB electrode materials. The potential organic polymers electrode materials for KIBs are typically identified by their functional groups that are electrochemically active for either oxidation or reduction [22,23]. The performance of the battery depends heavily on the electrolyte selection. A non-aqueous organic liquid electrolyte, which is based on various organic solvents, electrolyte salts, and additives, is the classic and most commonly used electrolyte due to the relatively high operating voltage window of potassium ion batteries [24,25]. In addition, due to their high safety, aqueous liquid electrolytes (which use water as a solvent), quasi-/solid-phase electrolytes, and ionic liquid electrolytes are also of concern [26,27]. Among these, polymer electrolytes are gaining a lot of attention since they combine the benefits of liquid electrolytes with inorganic solid electrolytes. Polymer binders are crucial to the battery system. The development of binders that can match with other components and produce KIBs with good performance is another crucial direction. The development and improvements in the use of polymers for KIBs, including PEMs, PEs, and binders, are summarized in this review article. To the best of our knowledge, the review of polymer-based applications in KIBs is the first to be reported, and each component is covered in detail. 

## 2. Polymer Electrode Materials (PEMs) for KIBs

Inorganic substances (such as layered metal oxides, metal sulfide phosphide, polyanionic compounds, carbon-based materials, and alloy-based materials) and organic substances are now used as electrode materials for KIBs [20,28,29,30,31,32,33,34,35]. Due to a number of benefits, organic polymer materials are more desirable than inorganic ones. First off, because organic polymer materials contain naturally plentiful C, H, N, O, and S, they are more environmentally benign, while transition metals are present in inorganic compounds. Second, materials made of organic polymers have adaptable molecular architectures that might make them better hosts for K-ions. Lastly, the fact that the feedstock is renewable makes polymers less expensive [36,37,38,39]. Polymers as electrode materials can be divided into cathode and anode materials for KIBs. Among them, research on polymer cathode materials is more extensive than on anode materials, with carbonyl (C=O groups) containing polymers making up the majority of the polymer cathode materials. 

### 2.1. Polymers as Cathode Materials for KIBs

If crystalline materials are utilized as cathodes, the bulky size of K^+^ would result in significant strain, which could be the biggest difficulty for KIBs. Currently, the majority of studies on cathode materials for KIBs are restricted to hexacyanometallates, layered oxides, polyanionic compounds, and organic compounds [40,41,42,43,44,45,46,47,48]. Figure 2 summarizes the advantages and disadvantages of these cathode materials for KIBs. More advanced cathode materials must be created, as the quest for cathode materials appropriate for KIBs has barely begun. Limited doping and increased dead mass are the two vital issues with using polymer as a KIBs electrode [23]. Therefore, there is a pressing need for research into developing novel polymers to address these issues. Carbonyl-containing polymers are in a class of cathode materials used in KIBs. Since poly (anthraquinone sulfide) (PAQS) has been employed in numerous studies as the cathode of LIBs, sodium-ion batteries, and magnesium-ion batteries and has demonstrated high capacity (~200 mAh/g) and good cycle stability [49,50,51], it is meaningful to examine the storage behavior of K^+^ in PAQS. Ji et al. [52] used a very simple method to synthesize PAQS (see Figure 3a for the unit molecular structure of PAQS) and investigate its electrochemical K^+^ storage performance for KIBs for the first time. The PAQS electrode exhibited a high reversible capacity of 190 mAh/g at the current density of 20 mA/g (Figure 3b), 84% of its theoretical capacity (for two K^+^ insertion/extraction). The PAQS electrode also displayed a good cycling performance, its capacity retention can reach 75% after 50 cycles, outperforming the organic molecular solid-PTCDA [53]. According to the CV results shown in Figure 3c, the K^+^ storage mechanism in PAQS was two K^+^ inserted into two sequential steps, which was similar to PAQS/Na cells [51]. A potential redox mechanism for potassium storage in PAQS is shown in Figure 3d. Another cathode polymer that has been mentioned in the previous report is poly (pentacenetetrone sulfide) (PPTS). Due to its large π-conjugated system, PPTS may achieve layer-by-layer stacking, effectively increasing the charge transport and ion diffusion of Na^+^ [54]. Based on this, Wang’s group [55] reported the PPTS as a cathode for KIBs with a superior rate capability and an ultralong cycling life. The PPTS delivered a reversible capacity of 260 mAh/g at a current density of 0.1 A/g over 100 cycles. It was impressive that the PPTS still exhibited a high reversible capacity of 190 mAh/g after 3000 cycles even at a high current density of 5 A/g, which was the best KIBs performances among all the reported polymer cathodes. The combination of a capacitive process, fast K^+^ diffusion, and low internal resistance may be responsible for this exceptional performance.

Based on the considerations of C=O groups, high electronic conductivity, and high redox potential, Jiang et al.’s group [56] prepared a series of polydiaminoanthraquinones (PQs) cathodes (PQ-1,5, PQ-1,4, and PQ-CN) for KIBs. The PQs with redox-active quinone-based segments are linked by a polyaniline skeleton. Due to the conjugated polymer backbones and excellent electrochemical reversibility of the functional groups, the PQs showed a high K storage capacity of 160~185 mAh/g. The PQ-CN performed electrochemically better than PQ-1,5 and PQ-1,4 in three PQs (having a greater reversible capacity of 184 mAh/g), which is due to the following two factors: (1) the PQ-CN has lower LUMO energy levels, allowing the electrode material to have a stronger electron affinity and a higher redox potential of 2.0 to 2.4 V; (2) the narrower LUMO-HOMO energy gap (E_g_) of PQ-CN is beneficial for the fast electron transport in the polymer electrode. 

To reveal the structure–performance relationship in KIBs, Tian et al. [57] synthesized polyimide (PI), polyquinoneimide (PQI), and conjugated microporous polymer (PI-CMP) as cathode materials in KIBs. The PQI exhibited the highest initial capacity, indicating that higher carbonyl group content was beneficial to increase the initial capacity, but the tight distribution of potassium carbonyl groups leads to rapid capacity decay. Moreover, similar to the conclusion of PPTS, the extended π-conjugation can improve the cycling stability and rate performance of the battery. Hu et al. [58] designed the polyimide@graphite nanosheets (PI@G) composite structure as a cathode for KIBs. The schematic diagram of the synthesis process of PI is presented in Figure 4a. The PI cathode stabilized with graphite nanosheets exhibited a maximal capacity of 142 mAh/g at 100 mA/g and maintains a capacity of 118 mAh/g over 500 cycles (Figure 4b,c), which may be a good guideline to design a high-performance cathode for KIBs. Zhang et al. [59] designed a porous polyimide@Ketjenblack (PIM@KB) structure via in situ polymerization as a cathode for fast rechargeable KIBs. The PIM@KB cathode delivered a high reversible capacity of 143 mAh/g at 100 mA/g and a reversible capacity of 105 mAh/g over 1000 cycles at 2 A/g. Furthermore, the possibility of superior rate capability (a capacity of ~106 mAh/g at a high current density of 3200 mA/g) of a full battery in Figure 4d offers a way to design polymer electrodes for KIBs with high capacity and rate capability. 

Some nitrogen-containing polymers have also been identified as potential cathode materials for KIBs, including poly (N, N′-diphenyl-p-phenylenediamine (PDPPD), poly (N-phenyl-5,10-dihydrophenazine) (p-DPPZ), and hexaazatriphenylene-based polymer (PHAT) [60,61,62]. Troshin et al. [60] synthesized and investigated redox-active PDPPD material for KIBs, which delivered a maximum capacity of 63 mAh/g and an 86% capacity retention after 500 cycles at 1C. However, ultrafast charge–discharge cycling of the PDPPD-based KIBs was unsatisfactory; this may be a significant restriction correlation between the battery operation at high current rates and the alkali-ion diffusion kinetics. In a later work, Troshin et al. [61] explored high-voltage p-DPPZ cathodes for KIBs for the first time. The p-DPPZ was prepared based on the reduction of phenazine to 5,10-dihydrophenazine and the following Buchwald–Hartwig condensation (Figure 5a). The p-DPPZ cathodes exhibited the highest capacity of 162 mAh/g at the current density of 200 mA/g (Figure 5b,c). At high current density of 2 A/g, the batteries exhibited maximum specific capacity of 131 mAh/g with a retention of 79 and 59% of this value after 1000 and 2000 cycles, respectively (Figure 5d). Additionally, Kapaev et al. [62] synthesized PHAT cathodes and showed comparable and very promising performances in Li-, Na-, and K-ion cells. The best reported performance was shown by the developed polymer PHAT in KIBs, which had high specific capacities, high energy densities, and great rate capabilities. Most remarkably, no capacity fading was observed after 4600 cycles at the high current density of 10 A/g. Their report further advanced the development of polymer cathodes for KIBs. We list the polymer cathodes for KIBs in Table 1.

### 2.2. Polymers as Anode Materials for KIBs

The use of polymers as anode materials for potassium ion batteries is relatively limited, with only little research attention. The team led by Zhang et al. [63,64] designed various conjugated polymer-based KIB anodes. They prepared π-conjugated polypyrene nanoflowers formed with nanosheets by oxidation polymerization [63]. The conjugated polypyrene polymer exhibited a stable reversible capacity of 302 mAh/g at 100 mA/g after 60 cycles, and a reversible capacity of 190 mAh/g even at a higher current density of 500 mA/g. Even after 30 days, polypyrene has very limited solubility in the conventional electrolyte of KIBs compared to pyrene (Figure 6a–d). According to the DFT calculations, polypyrene has a smaller HOMO-LUMO energy gap, which means higher electronic conductivity and faster redox kinetics after the polymerization of pyrene (Figure 6e). Therefore, the excellent electrochemical performance was attributed to the limited solubility against conventional electrolytes of KIBs, the enhanced electronic conductivity, and a larger K^+^ diffusion coefficient (10^−11^ ~ 10^−9^ cm^2^ s^−1^) (Figure 6f). Soon afterward, Zhang et al. [64] constructed a composite structure of polybenzo [1,2-b:4,5-b’]dithiophene-4,8-dione/reduced graphene oxide (PBDTO/rGO) as anode for KIBs. The PBDTO is a coaxial spiral structure conjugated polymer (Figure 6g), and the introduction of rGO drastically reduced the solubility of PBDTO in the electrolyte and enabled PBDTO/rGO to have high conductivity, low solubility, and high capacity. The PBDTO/rGO displayed an excellent KIBs performance with a reversible capacity of 395 mAh/g at 100 mA/g and a capacity of 200 mAh/g after 1000 cycles at the current density of 1 A/g (Figure 6h,i). Their works provided a reliable idea for designing environmentally friendly, high-performing, and sustainable PEMs for KIBs.

## 3. Polymer Electrolytes (PEs) for KIBs

Important battery system characteristics such as energy density, power density, cycle life, internal resistance, thermal stability, and safety are significantly influenced by electrolytes [65,66]. For instance, a battery’s internal resistance is significantly influenced by the ionic conductivity of the electrolyte [66,67]; it further impacts both the power and energy densities [68]. Moreover, the potential window of electrolyte also considerably affects the energy density and power density of KIBs by achieving higher cell potential [66]. For KIBs, although better fluidity of liquid electrolytic solutions allows for faster ion transfer and better interfacial penetration, ensuring contact between the electrode and electrolyte, conventional liquid organic electrolytes pose more serious safety problems, which hindered their practical applications [69]. Since Wright et al. [70] first developed the polymer electrolytes (PEs) composed of polyethylene and alkali metal salt in 1973, people discovered that the polymer electrolyte combined the advantages of liquid electrolytes and inorganic solid phase electrolytes, thus showing relatively good performance. High-performance PEs for LIBs and SIBs have also been the subject of extensive research, primarily concentrating on the mechanism of ion transport and the interface between electrolyte and electrode [71,72,73,74,75]. Researchers are spending more time on K^+^-conductive PEs (KCPEs), which are based on KIBs, as a result of the increased attention being given to KIBs. Many polymer-based electrolytes have been documented in numerous scientific publications up to this point, such as polyethylene oxide (PEO) [76,77,78,79,80,81]. However, there are few studies on KIBs based on PEs. In this section, we summarized the progress of PEs for KIBs by classifying the existing KCPEs into PEO-based PEs and non-PEO-based PEs.

### 3.1. PEO-Based PEs

PEO is widely used in PEs, due to its exceptionally high ability to dissolve/complex large concentrations of a wide variety of salts, good machining, and mechanical properties [82,83]. PEO-base PEs can be divided into two groups depending on whether the PEs include a single polymer or not. According to whether they are made of a single polymer PEO matrix, PEO-based PEs can be divided into single PEO-based PEs and non-single PEO-based PEs. We discuss each of these categories below. 

#### 3.1.1. Single PEO-Based Pes

The proper addition of potassium salt components to single PEO-based Pes is an efficient way to increase the polymer’s amorphous percentage. The benefits of K^+^ ion salt are its abundance, low cost, and less susceptibility to moisture [84]. By dissolving the suitable salt, the polymer’s structure and characteristics can also be altered [85]. In 2015, Chandra [86] reported the synthesis, ion conduction, and polymeric battery fabrication of K^+^-conductive solid Pes (SPEs): (1-x) PEO:x KBr (where 0 < x < 50 in wt%) film by the solvent-free/hot-press method. The conductivity of various SPEs films at room temperature is discovered to be dependent on the salt concentration, as illustrated in Figure 7a. The ionic conductivity (σ) of SPE film that increased with the addition or complexation of KBr salt, and the maximum conductivity of ~ 5.01 × 10^−7^ S/cm, was observed in SPE film with the KBr concentration of 30 wt% (70PEO:30KBr), which displayed an enhancement of two orders of magnitude compared to that of pure PEO. The reason for the increase in σ is due to the rises in ionic mobility (μ) and mobile ion concentration (n), as shown in Figure 7b. The increases in μ and n are mainly due to the crystallinity of the polymer decreasing or the availability of more conducting paths and the increasingly larger number of mobile ions, respectively. As the concentration rises, the conductivity tends to decline, possibly as a result of ion binding [87]. Additionally, when the mass ratio of the KBr concentration is higher than 50, the SPE films become fragile and less flexible. Moreover, Chandra et al. [88] reported K^+^ ion-conducting SPEs: (1-x) PEO: xKCl, where 0< x < 50 in wt%, and investigated the electrical properties of these newly synthesized SPEs. Similarly, in Kesharwani et al.’s work [89], they examined how different CH_3_COOK concentrations affected the conductivity of SPE films as well as the ion transport behavior in K^+^-ion conducting SPE films ((1-x) PEO: x potassium acetate (CH_3_COOK)). The SPE films: ((1-x) PEO: xCH_3_COOK) were prepared via hot-press technique and exhibited a relatively higher room temperature conductivity of ~2.74 × 10^−7^ S/cm when the PEO/CH_3_COOK weight ratio was 95:5.

This phenomenon indicates that the addition of different salts may have different effects on the same polymer host. Previous research has shown that metal salts with bigger anions have lower lattice energy and better solubility in common organic solvents, which is advantageous for improving ionic conductivity [90]. Based on this, Feng’s group [75] constructed all-solid-state KIBs with a PEO-bis(flourosulfonyl)imide potassium (PEO-KFSI)-based SPE; the used KFSI salt has large anionic groups. The impact of the (EO)/K^+^ molar ratio on the ionic conductivities of the PEO-KFSI SPE at 40 °C is shown in Figure 7c. Due to the addition of charge transfer and ion pairing, respectively, the ionic conductivity initially improves and subsequently begins to decrease when the molar ratio of (EO)/K^+^ grows. When the molar ratio of (EO)/K^+^ is 10, the ionic conductivity has a peak of 1.14 × 10^−5^ S/cm. The comparison of cycling performance of KIBs using a Ni_3_S_2_@Ni electrode between the PEO-KFSI-based SPE and the organic–liquid electrolyte (OLE) at a current density of 25 mAh g^−1^ was studied (Figure 7d). KIBs with SPE have an initial discharge capacity of 312 mAh g^−1^, which is lower than KIBs with OLE (381 mAh g^−1^), but the cell with SPE has good capacity retention ability, whereas the cell with OLE has enormous capacity fading and only achieves a low discharge capacity of 24 mAh g^−1^ after 20 cycles, which may be attributed to the high dissolubility of polysulfides into OLE. Recently, Elmanzalawy et al. [91] synthesized (PEO)*_n_*/potassium tetraphenylborate (KBPh_4_) (5 ≤ *n* ≤50) PE films by a solvent-free hot-pressing route (Figure 8a). They thoroughly and methodically investigated the ionic conductivity of (PEO)*_n_*/KBPh_4_ PEs at various temperatures and salt concentrations (Figure 8b–d). In the low-temperature regime below the melting point of PEO, PEO_15_ showed the highest conductivity of 1.06 × 10^−4^ S/cm at 55 °C and 7.43 × 10^−7^ S/cm at an ambient temperature of 25 °C; meanwhile, in the high-temperature above the melting point of PEO, the PEO_30_ showed the maximum conductivity (1.8 × 10^−3^ S/cm at 80 °C), although the salt concentration is low. Their findings highlighted the parameters affecting ionic conductivity both above and below the melting point of the crystalline PEO domains.

In addition to increasing the ionic conductivity of KCPEs by adding a single potassium salt to the PEO-based polymer host, the effect of increasing the conductivity can be achieved by adding additional inorganic fillers. Inorganic fillers can be divided into active and passive fillers according to their contributions. Active fillers participate in the transport of K^+^ and promote ionic conductivity. For passive fillers (Al_2_O_3_, SiO_2_, and TiO_2_, etc.), it has been discovered in earlier studies that introducing some 0D nanoparticles such as Al_2_O_3_, SiO_2_, TiO_2_, and ZrO_2_ into the polymer matrix could improve the ionic conductivity by 1–2 orders of magnitude; this is because adding these fillers can reduce the crystallinity of PEO and facilitate the motion of the polymer chains [92,93,94,95]. De et al. [96] synthesized PEO complexed with potassium iodide (KI) to investigate the ionic conductivity of alkaline-based PEs, and further studied the effect of ceria (CeO_2_) nanoparticle (size ≈ 10 nm) fillers on ionic conductivity in PEO/KI PEs, maintaining the mass ratio of PEO/KI at 80:20 and 85:15. The image of pure PEO (Figure 9a) showed a regular crystallized network, which is the evidence of a semicrystalline nature of pure PEO. The degree of crystallinity was reduced after the addition of KI salt as shown in Figure 9b. With further introduction of the CeO_2_ into PEO/KI, the morphology of host PEs showed dramatic changes, and PEO/KI/CeO_2_ composite PEs exhibited a flat surface when 20 wt.% of CeO_2_ nanoparticles were added (Figure 9c). Figure 9d presents the variation of conductivity with KI salt concentrations in PEO PEs complex at room temperature (RT); it is obvious that the conductivity of PEs increased gradually with increases in KI concentration, and the conductivity reaches the value of 1.53 × 10^−5^ S/cm at RT with 20 wt.% of KI salt. The variation of conductivity with CeO_2_ concentrations in the PEO/KI/CeO_2_ composite PEs (keeping the wt.% ratio of PEO to KI as 85:15 and 80:20) at RT are shown in Figure 9e, and the temperature dependence conductivity plots of different PEO/KI/CeO_2_ PEs are depicted in Figure 9f. The plot in Figure 9e displays two conductivity maxima at around 5 and 20 wt.% of CeO_2_, which is similar to many reported works on the existence of two maxima in composite PEs [97,98,99,100]. For CeO_2_ content of less than 5 wt.%, the percolating conduction path around nano-sized CeO_2_ and the trans arrangements of PEO are formed, which could enhance the ionic conductivity of the composite PEs for low content of CeO_2_. When the CeO_2_ concentration increases from 5 to 10 wt%, the Lewis acid–base interaction between the polar surface groups of CeO_2_ and the reduction of segmental motion resulted in a decrease in ionic conductivity. Above 10 wt.%, CeO_2_ interacts with ether oxygen of solvating K^+^, which can weaken the binding of K^+^ and enhance the conductivity. The highest value of conductivity was 2.15 × 10^−3^ S/cm when 20 wt% CeO_2_ was added to PEs. Furthermore, Chandra [101,102] applied another passive filler (SiO_2_) in the SPEs of 70PEO:30KBr to prepare a new KCPEs of (1-x) [70PEO:30KBr] + x SiO_2_ (0< x < 20 wt%); the SPEs were used as a polymer host and the SiO_2_ nano-particles (~8 nm) as a second-phase dispersoid. The conductivity of the polymer-salt/nano-filler SiO_2_ complexation system of [95(70PEO:30KBr):5SiO_2_] presented two orders of enhancement compared with (70PEO:30KBr), and delivered a maximum conductivity of 2.5 × 10^−5^ S/cm due to a consequence of dispersal of SiO_2_ in the SPE host that could promote ion migration. Agrawal et al. [103] prepared PEO-based nano-composite PE membranes: (70PEO: 30KNO_3_) + x SiO_2_, where x = 0, 1, 2, 3, 5, 8, 10, 12 wt. (%). Their research further proved that the fractional dispersal of nano-SiO_2_ into SPEs host: (70PEO: 30 KNO_3_) could enhance the RT conductivity by more than three-fold.

Agrawal et al. [104] also investigated the impact of KI as a second-phase active filler on room temperature conductivity (σ_rt_) of two first-phase KCPE films: [95PEO: 5KNO_3_] and [70PEO: 30KNO_3_]. In salt concentration-dependent conductivity of SPE films: [(1-x) PEO: xKNO_3_] research, two PEO/KNO_3_ SPE films with the mass ratio of PEO: KNO_3_ at 95:5 and 70:30 exhibited higher σ_rt_ of ~2.76 × 10^−7^ S/cm and ~4.31 × 10^−7^ S/cm, respectively (Figure 9g). The KI concentration-dependent conductivity variation study as shown in Figure 9h indicated that [(95PEO + 5KNO_3_) + 7 KI] and [(70PEO + 30KNO_3_) + 10 KI] composite PEs have relatively higher σ_rt_ ~6.15 × 10^−6^ and ~3.98 × 10^−6^ S/cm, respectively, which are three orders of magnitude higher than the σ_rt_ of pure PEO (~3.16 × 10^−9^ S/cm). Two maxima are present on the curves of Figure 8h at two distinct KI concentrations; this phenomenon has previously been observed in other composite PEs systems above, and its cause has also been identified [96].

#### 3.1.2. Non-Single PEO-Based PEs

The low ionic conductivity of the SPE based on PEO is mainly due to the high crystallinity and the simultaneous migration of cations and anions [105]. To improve ionic conductivity, multiple methods have been put forward to meet the practical performance requirements of PEs for KIBs, such as blending and copolymerization [106,107,108]. Blending is achieved by mixing two or more polymers, which is one of the effective methods used to decrease crystallinity [107]. The blend-based PEs have the advantages of a facile preparation process and easy control of physical properties by adjusting the proportion of polymers [108]. Some studies have been implemented on the blending of KCPEs systems. Poly(vinyl alcohol) (PVA) can assist in the formation of polymer blends, because it contains a carbon chain backbone with hydroxyl (-OH) groups (a source of hydrogen bonding) attached to methane carbons [109]. Based on this, Sarada et al. [109] prepared PEO-PVA blended PEs complexed with KIO_3_ salt. With rising temperature and dopant concentration, the (PEO + PVA)/KIO_3_ polymer blend electrolyte film electrical conductivity increased, and the maximum ionic conductivity was 4.77 × 10^−6^ S/cm at 308 K when the mass ratio of PEO/PVA/KIO_3_ was 35:35:30. In Chen’s group [107,108], they prepared K^+^ ion-conducting polymer blend electrolyte PVC/PEO: KCl SPEs and PVC + PEO + KBr SPEs, respectively, and conducted performance research. In the PVC/PEO: KCl systems, the conductivity value is significantly increased with an increase in KCl salt concentration. The maximum value of ionic conductivity (8.29 × 10^−6^ S/cm) was obtained at the PEO/PVC/KCl weight ratio of 42.5:42.5:15 at 303 K, which was higher than that of pure PVC/PEO (3.09 × 10^−6^ S/cm). The results of solid polymer blend electrolyte PVC + PEO + KBr systems were similar. The highest ionic conductivity value of PVC + PEO + KBr (42.5:42.5:15) polymer blend electrolyte can reach 2.56 × 10^−5^ S/cm at ambient temperature. These polymer blend electrolyte systems’ increased ionic conductivity over single PEO-based PEs indicated that blending is a successful method for enhancing KCPEs’ ionic conductivity. Table 2 presents the all PEO-based PEs’ performance metrics for KIBs.

### 3.2. Non-PEO-Based PEs

In addition to PEO-based PEs, several efforts have been undertaken to build efficient PEs systems using a variety of different polymers, including poly(vinyl alcohol) (PVA), polyvinyl pyrrolidone (PVP), poly (methyl methacrylate) (PMMA), polyacrylonitrile (PAN), etc. [109,110,111,112,113,114,115,116,117,118,119,120,121,122,123,124]. The performance parameters of non-PEO-based PEs for KIBs are shown in Table 3. Ravi and Narasimha Rao et al. [85,110] prepared KCPEs films based on PVA hosts complexed with KCl and KBr salts separately and studied the related parameters of the assembled electrochemical cell. PVA was selected as the host polymer due to its excellent chemical resistance and ease of processing to form films [111] and good charge storage capacity. In PVA/KCl systems, the KCPEs with different weight ratios of PVA and KCl (pure PVA, 95:05, 90:10, and 85:15) were prepared [110]. The conductivity values of the PEs at room temperature increased with the increase in dopant concentration. Similarly, in the PE systems of PVA complexed with 5, 10, and 15 wt% KBr salt, the ionic conductivity increased with the increase in dopant concentration as well as temperature. The (PVA:KBr) (85:15) KCPE has the highest ionic conductivity of 1.23 × 10^−5^ S/cm at 303 K, which was improved by three orders of magnitude compared with pure PVA. PVP is also a typical polymer host for KCPEs. Kumar et al. [112] reported on (PVP + KIO_3_) PEs systems. The conductivity of PEs increased sharply to ~1 × 10^−9^ S/cm on complexing with 10 wt.% of KIO_3_ at RT, demonstrating poor ionic conductivity despite being much higher than pure PVP conductivity (~10^−13^ S/cm), and PVP-based PE could promote the dissociation of metal salts. This could be solved by polymer blending; Sharma’s group [113] designed the PVP/PVA/KIO_3_ (35:35:30 in wt.%) with ionic conductivity of 1.22 × 10^−5^ S/cm at RT, which was greatly improved compared with the work of (PVP + KIO_3_) with ionic conductivity of only 1 × 10^−9^ S/cm. Synthesizing copolymers is also an effective method to improve ionic conductivity. Two or more chains with various chemical characteristics make up the copolymers. A local separation structure at the nanoscale is produced by the net repulsive force between the segments and offers excellent control over ion transport and the mechanical properties of the film [114]. Kim et al. [114] prepared a graft copolymer consisting of a poly(epichlorohydrin) (PECH) main chain (60 wt.%) and poly (oligo (oxyethylene) methacrylate) (POEM) side chains (40 wt.%) by atom transfer radical polymerization. The PE was prepared via KI salt added to the PECH-g-POEM copolymer, and the highest ionic conductivity reached 3.7 × 10^−5^ S/cm at RT at 5 wt.% of salt concentration. The PEs’ excellent electrochemical performance suggests that the copolymerization process has a lot of potential.

The PEs matrix category also includes polycarbonate. Due to its environmental friendliness, the new polycarbonate Poly (propylene carbonate) (PPC) has garnered considerable interest [115,116]. Inspired by the work of Cui’s group [115] on SPE, they used PPC as the ionic transit material and developed a cellulose nonwoven membrane as the backbone for LIBs. Feng et al. [117] designed a PPC-KFSI with cellulose nonwoven backbone SPE (PPCB-SPE) for high-safety solid-state KIBs, as shown in Figure 10a. The PPCB-SPE presented the maximum ionic conductivity of 1.36 × 10^−5^ S/cm at a KFSI salt concentration of 18 wt% at 20 °C (Figure 10b). Additionally, PPCB-SPE has an ionic conductivity of up to 6.06 × 10^−5^ S/cm at 120 °C, which could be attributed to the decreased T_g_ value (Figure 10c). Figure 10d shows that the PPCB-SPE had an excellent electrochemical stable window of 4.15 V, which was enough to meet the demands of the perylene-3,4,9,10-tetracarboxylic dianhydride (PTCDA) cathode electrode for KIBs. The assembled full cell (K||PPCB-KFSI SPE||PTCDA) exhibited an excellent cycle performance (Figure 10e). The discharge capacity of the PTCDA cathode with PPCB-KFSI SPE reached 113 mAh g^−1^ at 20 mA g^−1^ after an activation process, and the capacity retention rate can reach 84.3% after 40 cycles, while the capacity with conventional organic–liquid electrolyte (OLE) suffered a great fading of 45.7%. The rapid fading of the capacity with OLE may be on account of the high solubility of PTCDA into the OLE (the inserted digital image of Figure 10e), while the SPE could effectively inhibit the dissolution of PTCDA and other irreversible side reactions [118].

Polyurethane (PU) is a linearly segmented block copolymer with a multiphase polymeric chain structure consisting of soft segments and hard segments. The soft segment originated from polyols and can as a polymeric solvent to dissolve cations, the hard segment is contributed by the isocyanates and can be functionalized to improve electrochemical stability to permit the fabrication of the PEs [119,120]. Considering that the PU host with N-H, C=O, and C-O-C groups can coordinate with K^+^, Aung et al. [121] prepared a jatropha oil-based polyurethane acrylate (PUA) gel PE (GPE), the PUA incorporated with 25 wt.% KI salt has an optimal conductivity of 1.59 × 10^−4^ S/cm at RT. However, the electrochemical stability window of this gel PE was only 2 V, which limited the selection of the cathode and the energy density of the full battery. Besides, the other GPEs based on polyacrylonitrile (PAN) and poly(methyl methacrylate) (PMMA) have been studied [122,123]. In PAN-based GPE, Kumar et al. [122] added EC as the plasticizer into the PAN matrix to prepare a K^+^-based GPE membrane with improved ionic conductivity, the maximum conductivity of PAN:KI (70:30 in wt.%) was 2.089 × 10^−5^ S/cm at 30 °C, which is nine-orders greater than that of pure PAN (<10^−14^ S/cm). The addition of plasticizing solvent EC also improved the ionic conductivity, which was due to an increased number of charge carriers by the larger dissolution of the electrolyte salt.

## 4. Polymer Binders for KIBs

As a component of the battery system, trace binders play an important role. The main functions of the binder include (1) As a dispersant or thickening agent to improve the uniformity of electrode components; (2) Bond the active substance, conductive agents, and fluid collector to maintain the integrity of the electrode structure; (3) To provide the required electron conduction in the electrode; (4) Improve electrolytic wettability and promote ion transport at the electrode-electrolyte interface [124,125,126].

Since KIBs work in a similar way to LIBs, most of the binders used in LIBs can be directly used in KIBs, such as polyvinylidene fluoride (PVDF), and so on [127]. Binders include natural binders and artificial binders, among which artificial polymer binders are the most widely used in batteries. Among the artificial binders, PVDF is the most used in LIBs; other artificial polymer adhesives include PVA, poly(acrylic acid) (PAA), polytetrafluoroethylene (PTFE), polyethylenimine (PEI), and other polymers [128,129,130,131,132,133,134]. As for the classification of binders, in addition to man-made binders and natural binders, there are oily binders and water-based binders, fluorinated binders and non-fluorinated binders, reactive binders and non-reactive binders, and non-reactive binders are the main binders in all kinds of energy storage batteries at present [124].

For different electrode materials, the selected binder is not the same. For the cathode material, most of the selected binders are oil-soluble binder PVDF. Water-soluble binders are rarely used [40,135,136,137,138,139,140,141,142,143]. This is because the cathode material is more suitable for the selection of an oily binder [144]. For the anode material, the choice of binder is relatively more kind (Table 4) [145,146,147,148,149,150,151,152,153,154,155]. In previous studies, it was found that the performance of Li/Na ion batteries with non-PVDF binder was improved [156,157]. Because anode electrodes for KIBs generally exhibit a large volume change and SEI instability, the cycle life of KIBs can be effectively improved by choosing the appropriate binders instead of traditional PVDF binders. Based on these studies, Mai’s group [155] studied the effects of different binders (CMC and PVDF binder) on the electrochemical performance of porous hollow carbon sub microspheres (HCS) for KIBs, the results show that the HCS electrode using CMC as a binder exhibited more excellent reversible capacity and stable cycle life compared with the PVDF, as shown in Figure 11. The initial Coulombic efficiency (ICE) of HCS-600 using CMC binder is 52%, which is significantly better than HSC-600 using PVDF binder (26%) (Figure 11 g). The cycling performance of HSC-600 using CMC binder is significantly better than HSC-600 using PVDF binder. At the current density of 50 mA/g, the capacities of HSC-600 using CMC and PVDF binders are 211 and 89 mAh g^−1^ after 50 cycles, respectively. Even at a larger current density of 1 A/g, the capacity of HSC-600 using CMC binder can reach 111 mAh g^−1^ after 3000 cycles, which is significantly better than HSC-600 using PVDF binder (only 18 mAh g^−1^ after 3000 cycles) (Figure 11h). Cao et al. [154] also studied the effects of three different binders (PVDF, CMC, and CMC/SBR) on the electrochemical performance of the SnS_2_/graphene nanocomposite anode based on the KIBs. The results show that the electrode with CMC/SBR hybrid binder exhibits better capacity retention and better cycle stability. In addition, in the latest work of Yang’s research group [158], he discussed binder chemistry in Na/K ion batteries. The effects of water-soluble CMC binder and oil-soluble PVDF binder on the performance of KIBs with several different electrode materials (including intercalation, conversion, and alloy anode) were systematically investigated. Figure 12 displays the advantages of CMC over PVdF as the binder in different electrode materials (TiO_2_, hard carbon, MoS_2_, and Bi/C). Although binders play a critically important role in battery systems, there is little research work on newer and more efficient binders for KIBs. In addition to providing good adhesion, mechanically resilient, and electrical contact between the active material and the current collector, an effective binder must also ensure electrochemical stability and facilitate the formation of a thin and stable solid electrolyte interface (SEI). It can be seen from Table 4 that fluoropolymers and water-soluble cellulose/gum derivatives are only used as binders in KIBs in most works, of which PVDF is still the main component. More recent work has also focused on other polymers such as PAA, SBR, etc. [148,152,159]. In the latest work by Gribble et al. [160], to enhance the conductivity of the binder, adhesion and electrical contact are used to increase the charging speed and to mitigate the capacity decay mechanism. To this end, they chose the electronically conductive polymer mixture PETOD:PSS as the binder for KIBs (Figure 13a). In their study, the electrochemical, mechanical, and thermal safety properties of PEDOT:PSS as a binder were comprehensively explored. The addition of carbon black (CB) to continuously conducting PEDOT:PSS/CB binder reduces the electrical insulation of “dead” graphite. Compared to PVDF/CB, the PEDOT:PSS/CB graphite anodes display greater capacity, cycle lifetime, rate capability, and reduced impedance due to enhanced electronic properties (Figure 13b). Thermal safety is crucial for any new battery technology to enter the market; thus, the DSC of circulating electrodes was investigated. Compared with the electrode using PVDF binder, the thermal runaway of the PEDOT:PSS electrode was delayed due to greater stability against reduced potassium, the heating started at 162 °C, and the heat release was reduced to 567 from 720 J g^−1^ (Figure 13c). Figure 13d provides the supporting evidence for enhanced stability of PEDOT:PSS against reduced potassium, which displayed DSC profiles for PEDOT:PSS and PVDF polymer films against K metal. PVDF undergoes dehydrofluorination at 175 °C; in comparison, PEDOT:PSS appears stable against K metal until 230 °C. Their results indicated that the PEDOT:PSS with low surface area, the poor wettability, and greater stability against K^0^ may help to improve anode thermal stability of KIBs. In addition, tailoring the SEI through binder modifications and electrolytes may be effective toward developing safer KIBs.

## 5. Conclusions

The application of organic polymers in KIBs has shown great potential. In this review, we have comprehensively summarized the development of a series of polymers as different components in the KIBs. Polymer cathode materials mainly contain -C=O group polymers and -nitrogen polymers. Among them, the construction of polymer cathodes containing C=O groups, insoluble redox activity, high electron conductivity, and high redox potential is of great significance for the development of new high-performance cathode material. As for the polymer’s anode used in KIBs, it is mainly a conjugated polymer. K^+^-conductive PEO-based PEs overcome the shortcomings of traditional electrolytes and can greatly improve ionic conductivity by adding an appropriate potassium salt component and blending. In addition, the development of high-performance polymer binders is also worth investigating further. Overall, it should be admitted that the development of polymers in KIBs systems is still in its infancy, and with the improvement in polymer electrode materials and binders design, the exploration of compatible PEs, and the understanding of the basic principles of the electrochemical behavior of KIBs, KIBs will undoubtedly become a competitive and attractive application choice for large-scale energy storage systems in the future.

## Figures and Tables

**Figure 1 polymers-14-05538-f001:**
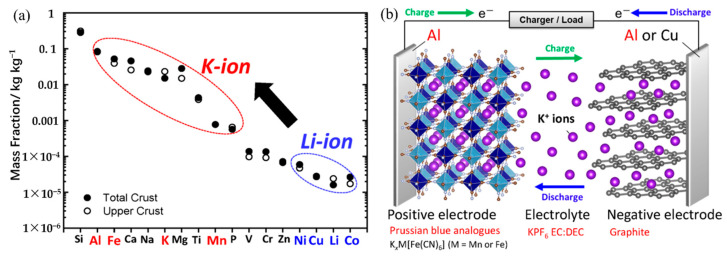
(**a**) Elemental abundance in the Earth’s crust. (**b**) Schematic illustration of the cell configuration and operational mechanism of a typical K-ion battery. (**a**,**b**) Reproduced with permission from Ref. [7]. Copyright 2020 American Chemical Society.

**Figure 2 polymers-14-05538-f002:**
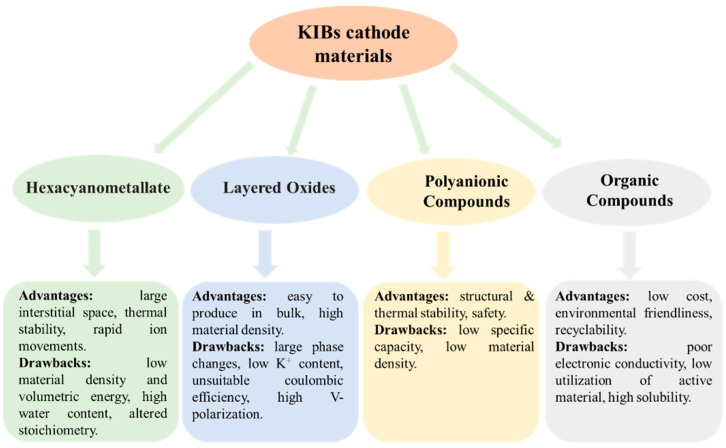
Advantages/drawbacks of different cathode systems for KIBs.

**Figure 3 polymers-14-05538-f003:**
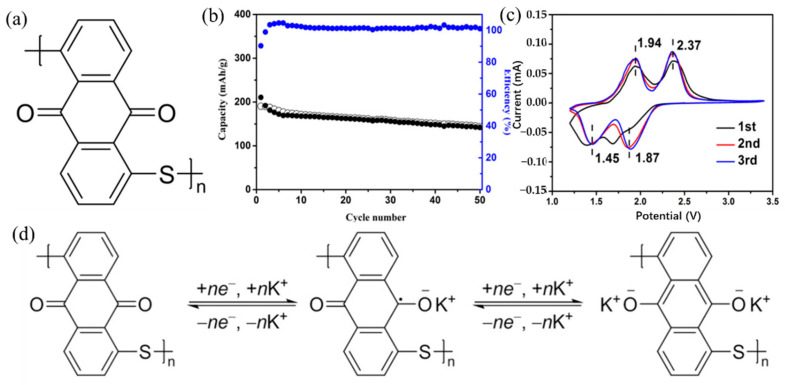
(**a**) Unit molecular structure of PAQS. (**b**) Cycling performance of PAQS/K cells between 1.5 and 3.4 V vs. K^+^/K at a current density of 20 mA/g. (**c**) CV curves of PAQS in KTFSI/DOL + DME for the initial three cycles between 1.2 and 3.4 V at a scan rate of 0.1 mV/s. (**d**) A possible redox mechanism of potassium storage in PAQS. (**a**–**d**) Reproduced with permission from Ref. [52]. Copyright 2016 Elsevier B.V.

**Figure 4 polymers-14-05538-f004:**
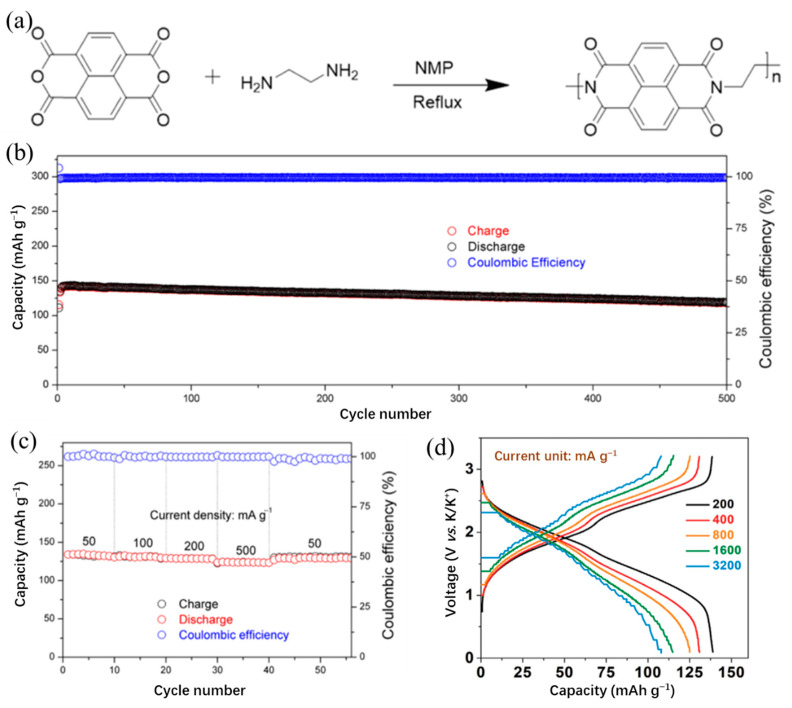
(**a**) Schematic diagram of synthesis process of PI. (**b**) Long-term cycling performance of PI@G for 500 cycles at 100 mA/g. (**c**) Rate performance of PI@G with current densities from 50 to 500 mA/g. (**a**–**c**) Reproduced with permission from Ref. [58]. Copyright 2019 American Chemical Society. (**d**) Charge profiles of the PIM@KB//NC full cell at various current densities. Reproduced with permission from Ref. [59]. Copyright 2020 Wiley-VCH GmbH.

**Figure 5 polymers-14-05538-f005:**
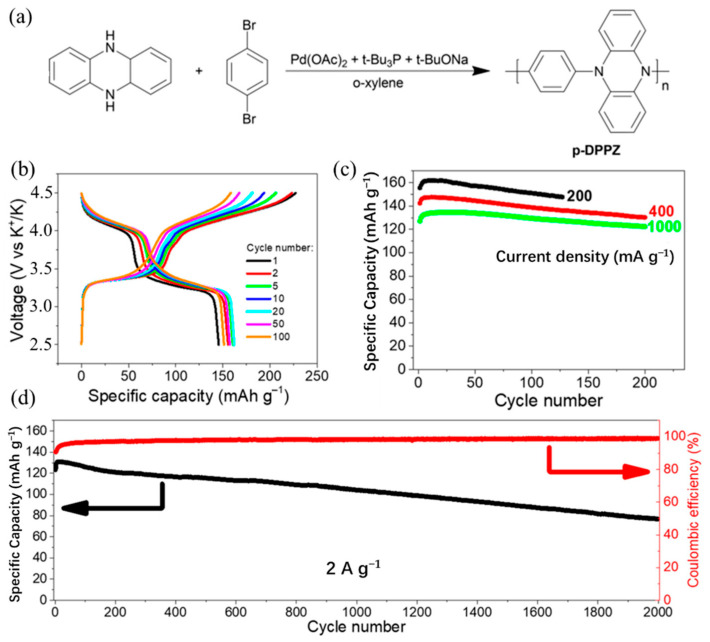
(**a**) The synthesis of p-DPPZ. (**b**) Charge and discharge characteristics of KIBs with p-DPPZ cathode and 2.2 M KPF_6_ in diglyme electrolyte cycled in the voltage range of 2.5−4.5 V at 200 mA/g current density. (**c**) Evolution of specific capacity of KIBs with p-DPPZ cathode under cycling at different current densities. (**d**) Long-term cycling stability of cell at current density of 2 A/g. (**a**–**d**) Reproduced with permission from Ref. [61]. Copyright 2019 American Chemical Society.

**Figure 6 polymers-14-05538-f006:**
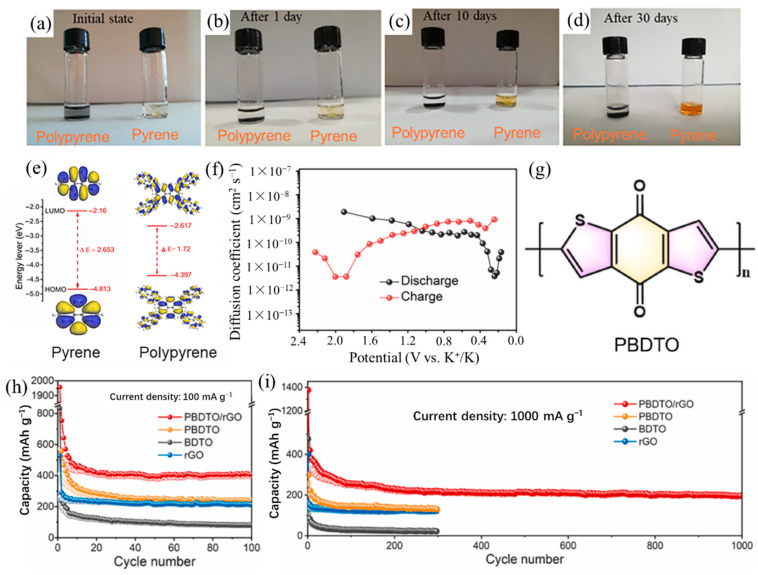
(**a**–**d**) Digital photos of the pyrene and polypyrene dispersed in the electrolyte of KIBs for different time intervals. (**e**) Theoretical calculated HOMO and LUMO in optimized structures of pyrene and polypyrene. (**f**) Diffusion coefficients at different potentials during potassiation and depotassiation of polypyrene. (**a**–**f**) Reproduced with permission from Ref. [63]. Copyright 2021 Elsevier B.V. (**g**) The molecular structure of PBDTO. Cycling performance (h) at 100 mA/g and (i) at 1000 mA/g of PBDTO/rGO, PBDTO, BDTO, and rGO. (**g**–**i**) Reproduced with permission from Ref. [64]. Copyright 2022 Elsevier B.V.

**Figure 7 polymers-14-05538-f007:**
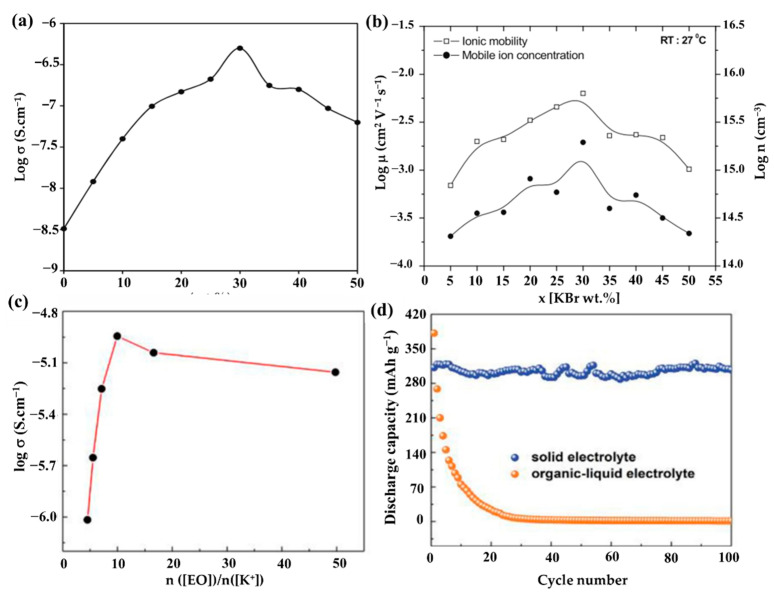
(**a**) “Log σ-x” plot for the hot-pressed SPEs: (1-x) PEO: x KBr, where x in wt%. (**b**) “Log μ-x” and “log n-x” plots for the hot-pressed SPEs: (1-x) PEO:x KBr. (**a**,**b**) Reproduced with permission from Ref. [86]. Copyright 2022 IACS. (**c**) Ionic conductivity of the PEO-KFSI SPE with varied molar ratio of [EO]/[K^+^] at 40 °C. (**d**) Comparison of cycling performance using 3 h Ni_3_S_2_@Ni electrode between PEO-based SPE and the organic–liquid electrolyte at a current density of 25 mAh/g. (**c**,**d**) Reproduced with permission from Ref. [75]. Copyright 2019 Elsevier B.V.

**Figure 8 polymers-14-05538-f008:**
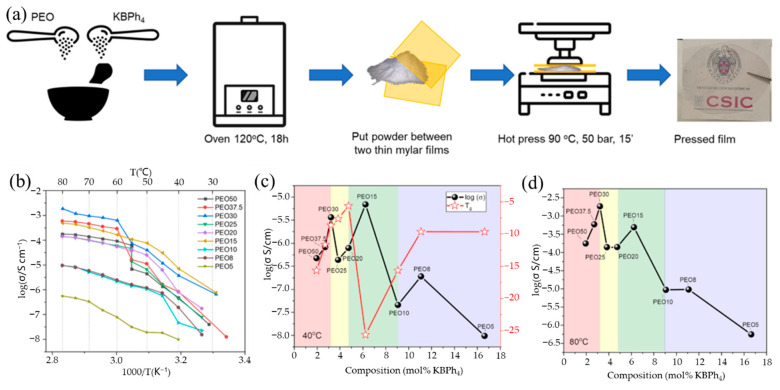
(**a**) Schematic sequence of the synthesis route of PEO/KBPh4 electrolyte films by hot pressing. (**b**) Arrhenius plots for all of the polymer/salt composite electrolytes. (**c**,**d**) Isotherms of ionic conductivity vs. mole fraction of KBPh_4_ in the polymer electrolyte at (**c**) below (40 °C) and (**d**) above (80 °C) the melting point of crystalline PEO. T_g_ values are plotted in the secondary y-axis of (**c**). (**a**–**d**) Reproduced with permission from Ref. [91]. Copyright 2022 American Chemical Society.

**Figure 9 polymers-14-05538-f009:**
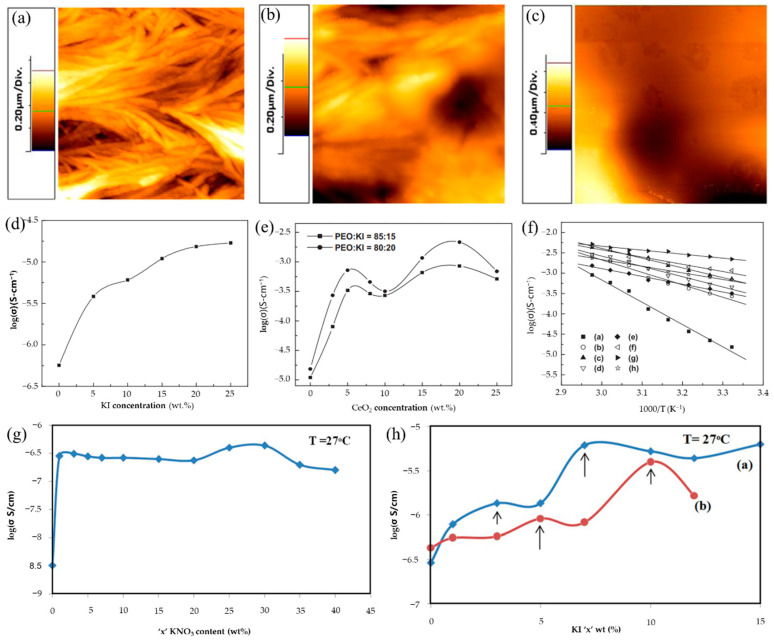
Two-dimensional AFM topographic images of (**a**) pure PEO; (**b**) PEO-KI (wt.% ratio 80:20) and (**c**) 20 wt.% CeO_2_ of PEO-KI-CeO_2_ composite polymer electrolytes. (**d**) Variation of conductivity with KI salt concentrations in polymer electrolyte complex at room temperature. (**e**) Variation of conductivity with CeO_2_ concentrations in the composite polymer electrolyte complex at room temperature. (**f**) Temperature dependence conductivity plots of PEO-KI-CeO_2_ composite polymer electrolytes with different CeO_2_ concentrations ((**a**–**h**): 0 wt.%−25 wt%). (**a**–**f**) Reproduced with permission from Ref. [96]. Copyright 2011 Elsevier Ltd. (**g**) Salt concentration-dependent conductivity variation for hot-press-cast SPE films: [(1-x) PEO: xKNO_3_]. (**h**) Filler (KI) concentration-dependent conductivity variation for CPE films: [(95PEO: 5KNO_3_) + KI] (♦); [(70PEO: 30KNO_3_) + KI] (•). (**g**,**h**) Reproduced with permission from Ref. [104]. Copyright 2017 Elsevier B.V.

**Figure 10 polymers-14-05538-f010:**
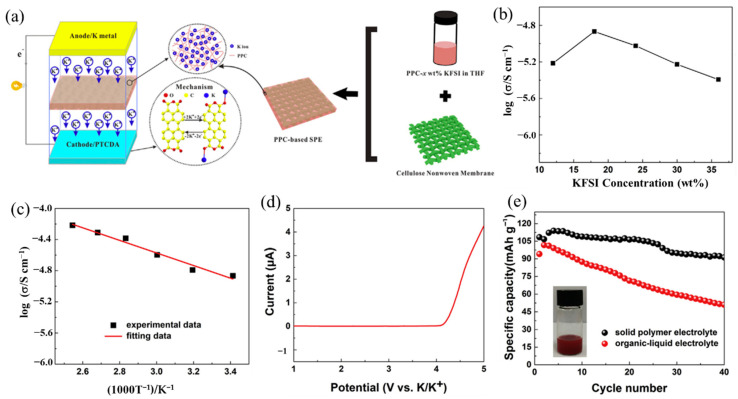
(**a**) The design of PPC-based solid polymer electrolyte with cellulose nonwoven backbone (PPCB-SPE) for high-safety solid-state KIBs. (**b**) Ionic conductivity of the SPE with varied concentration of KFSI in PPC. (**c**) Ionic conductivity of the SPE at different temperature with a concentration of PPCB-SPE-18 wt% KFSI. (**d**) The linear-sweep voltammetry results for the SPE. (**e**) Cycling performance of the assembled full cell with PPCB-SPE or organic–liquid electrolyte at a current density of 20 mA/g. The inserted digital image is the solubility test of PTCDA in 1M KFSI in EC/DEC (*v/v*, 1:1) electrolyte. (**a**–**e**) Reproduced with permission from Ref. [117]. Copyright 2018 Elsevier B.V.

**Figure 11 polymers-14-05538-f011:**
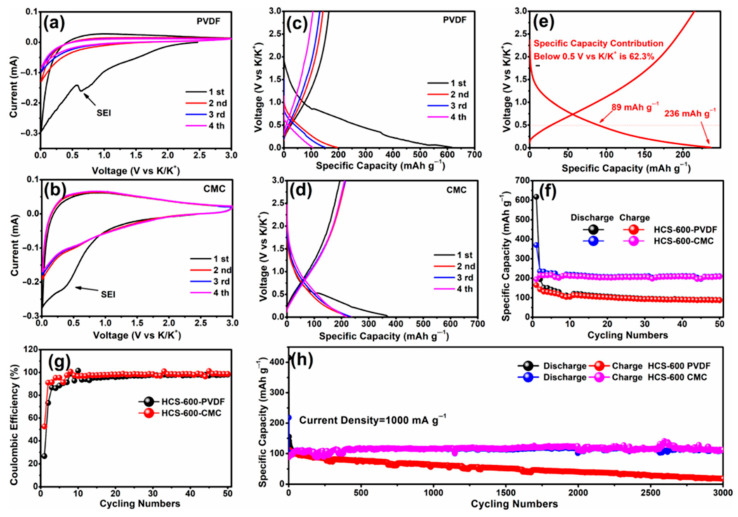
CV curves of HCS-600 using (**a**) PVDF and (**b**) CMC binders. Galvanostatic charge–discharge profiles of HCS-600 using (**c**) PVDF and (**d**) CMC binders. (**e**) Second discharge specific capacity contribution of HCS-600 using the CMC binder below 0.5 V vs. K/K^+^. (**f**) Cycling performance and (**g**) Coulombic efficiency of HCS-600 using PVDF and CMC binders tested at 50 mA/g. (**h**) Long-term cycling performance tested at 1000 mA/g. (**a**–**h**) Reproduced with permission from Ref. [155]. Copyright 2019 American Chemical Society.

**Figure 12 polymers-14-05538-f012:**
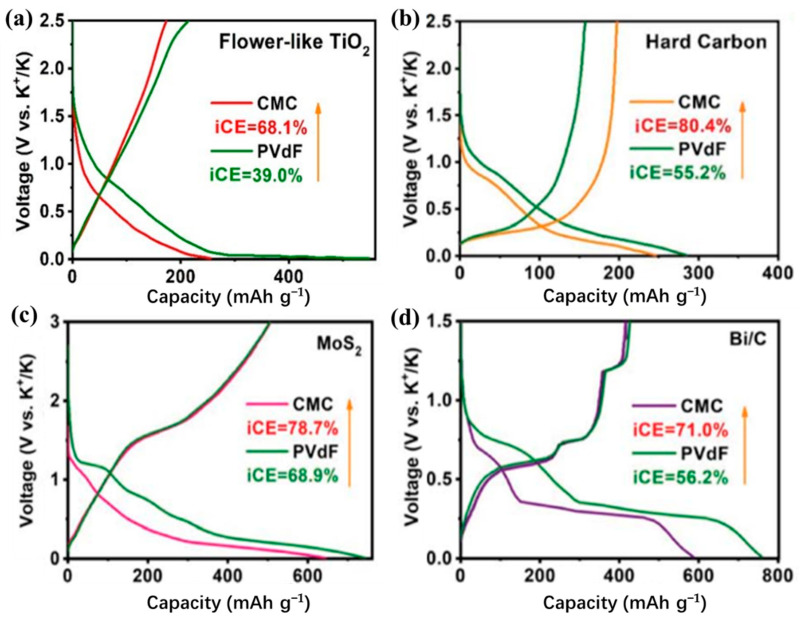
Advantages of CMC over PVdF as the binder in different electrode materials. (**a**) Flower-like TiO_2_, (**b**) hard carbon, (**c**) MoS_2_, and (**d**) Bi/C as the anode materials for K storage at 0.1 A/g. (**a**–**d**) Reproduced with permission from Ref. [158]. Copyright 2022 The Royal Society of Chemistry.

**Figure 13 polymers-14-05538-f013:**
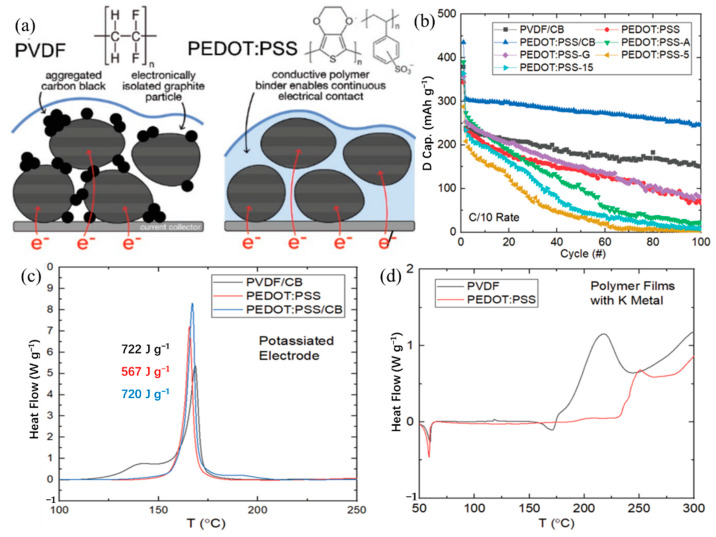
(**a**) Proposed mechanism for improving the performance of the graphite anode in a KIB. (**b**) The capacity performance of graphite half-cells using various binders in KFSI electrolyte with C/10 constant-current rate cycling. DSC profiles for cycled graphite anodes with varying binder compositions with (**c**) 100 SOC, (**d**) 0 SOC and wash. (**a**–**d**) Reproduced with permission from Ref. [160]. Copyright 2022 Wiley-VCH GmbH.

**Table 1 polymers-14-05538-t001:** The performance of polymer cathodes for KIBs.

Active Cathode Materials	Potential Window (V)	Reversible Capacity (mAh/g)	Current Density (mA/g)	Reference
PAQS	1.5–3.4	~190 (50 cycles)	20	[52]
PPTS	0.8–3.2	~190 (3000 cycles)	5000	[55]
PQs	1.2–3.2	~105 (200 cycles)	250	[56]
PI-CMP	1.5–3.5	~80 (1000 cycles)	1000	[57]
PI@G	1.4–3.5	~118 (500 cycles)	100	[58]
PIM@KB	1.5–3.4	~105 (1000 cycles)	2000	[59]
PDPPD	2.5–4.5	~54 (500 cycles)	1C	[60]
P-DPPZ	2.5–4.5	~79 (2000 cycles)	2000	[61]
PHAT	2–4.7	~175 (4600 cycles)	10,000	[62]

**Table 2 polymers-14-05538-t002:** Performance parameters of PEO-based Pes for KIBs.

Electrolyte Composition	Ionic Conductivity (S/cm)	References
PEO/KBr	5.0 × 10^−7^	[86]
PEO/KCl	5.6 × 10^−7^	[88]
PEO/CH_3_COOK	2.74 × 10^−7^	[89]
PEO/KFSI	2.74 × 10^−4^ at 60 °C	[75]
PEO/KBPh_4_	1.8 × 10^−3^ at 80 °C 1.1 × 10^−4^ at 55 °C	[91]
PEO/KI/CeO_2_	2.15 × 10^−3^	[96]
PEO/KBr/SiO_2_	2.5 × 10^−5^	[101,102]
PEO/KNO_3_/KI	6.15 × 10^−6^	[104]
PEO/KNO_3_/SiO_2_	1.07 × 10^−6^	[103]
PEO/PVA/KIO_3_	4.77 × 10^−6^	[109]
PEO/PVC/KBr	2.56 × 10^−5^	[108]
PEO/PVC/KCl	8.29 × 10^−6^	[107]

**Table 3 polymers-14-05538-t003:** Performance parameters of non-PEO-based PEs for KIBs.

Electrolyte Composition	Ionic Conductivity (S/cm)	Reference
PVA/KCl	9.68 × 10^−7^	[110]
PVA/KBr	1.23 × 10^−5^	[85]
PVP/KIO_3_	1 × 10^−9^	[112]
PVP/PVA/KIO_3_	1.22 × 10^−5^	[113]
PAN/KI/EC	2.089 × 10^−5^	[122]
PPCB-KFSI	1.36 × 10^−5^	[117]
PUA/KI	1.59 × 10^−4^	[121]
PECH-g-POEM/KI	3.7 × 10^−5^	[114]
PMMA/KPF6/EC:DEC:FEC	4.3 × 10^−3^	[123]

**Table 4 polymers-14-05538-t004:** Application of some representative binders in KIBs.

Active Materials	Binders	Reversible Capacity (mAh/g)	Current Density (A/g)	Reference
KVP (cathode)	PVDF	~70 (2500 cycles)	0.5	[135]
KMF (cathode)	PVDF	~136 (320 cycles) ~110 (2150 cycles) ~76.3 (6000 cycles)	0.03 0.1 0.5	[136]
KVPO_4_F@3DC (cathode)	PVDF	~90 (105 cycles) ~51.26 (550 cycles)	0.05 0.5	[137]
c-KMCNO (cathode)	PVDF	~42.6 (300 cycles)	0.1	[138]
KPBNPs (cathode)	PVDF	~73.2 (50 cycles)	0.05	[40]
KFeC_2_O_4_F (cathode)	PVDF	~105.3 (2000 cycles)	0.2	[139]
TBAPM (cathode)	CMC	~232 (16 cycles) ~110 (50 cycles)	0.02 0.1	[140]
MB (cathode)	PVDF	~139.5 (500 cycles) ~75 (4500 cycles)	0.1 2	[141]
K_0.5_MnO_2_ (cathode)	PVDF	~102 (50 cycles)	0.02	[142]
FeS_2_@CNBs (cathode)	PVDF	~221 (700 cycles)	0.1	[143]
MoS_2_@rGO (anode)	PVDF	~424.6 (1000 cycles)	0.5	[145]
SHCS (anode)	PVDF	~150 (1000 cycles)	3	[146]
S/N@C (anode)	CMC	~65 (900 cycles)	2	[147]
SnSb/C (anode)	CMC+PAA	~419 (600 cycles) ~340 (800 cycles)	0.05 1	[148]
Graphite (anode)	PAANa CMCNa PVDF	~231.1 (50 cycles) ~174.2 (50 cycles) ~168.6 (50 cycles)	0.05	[149]
CoPSe/NC (anode)	CMCNa	~317 (100 cycles) ~203 (2000 cycles)	0.1 5	[150]
MCOs (anode)	PVDF	~240 (100 cycles) ~100 (1300 cycles)	0.1 1	[151]
SC (anode)	CMCNa+SBR	~296 (50 cycles) ~200 (1000 cycles)	0.1C 1C	[152]
NCM (anode)	PVDF	~358.4 (100 cycles) ~189.5 (1800 cycles)	0.5 2	[153]
SnS_2_/graphene composite	PVDF CMC/SBR CMC	~150 (50 cycles) ~559 (50 cycles) ~458 (50 cycles)	0.1	[154]
HCS-600	CMC PVDF	~111 (3000 cycles) ~18 (3000 cycles)	1	[155]

Abbreviations: K_3_(VO)(HV_2_O_3_)(PO_4_)_2_(HPO_4_) (KVP). Potassium manganese hexacyanoferrate (K_2_Mn[Fe(CN)_6_]) (KMF). Amorphous carbon network-modified KVPO_4_F composite (KVPO_4_F@3DC). P_3_-type K_0.5_Mn_0.8_Co_0.1_Ni_0.1_O_2_ porous microcuboids (c-KMCNO). Potassium prussian blue K0._220_Fe[Fe(CN)_6_]_0.805_ 4.01H_2_O nanoparticles (KPBNPs). Tetra-n-butylammonium phosphomolybdate (TBAPM). Methylene blue (MB). FeSe_2_@N-doped carbon nanoboxes (FeSe_2_@C NBs). MoS_2_ anchored on reduced graphene oxide (MoS_2_@rGO). Sulfur-grafted hollow carbon spheres (SHCS). S and N co-doped thin carbon (S/N@C). Cobalt phosphoselenide (CoPSe) nanoparticles embedded in metal–organic framework (MOF)-derived N-doped carbon matrix (CoPSe/NC). Mesoporous carbon octahedrons (MCO). Pitch-derived soft carbon (SC). N-doped amorphous carbon/graphite-coupled polyhedral microframe (NCM). Porous hollow carbon spheres (HCS).

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
