# Peer review of "Recent Advances in Polymers for Potassium Ion Batteries"

_polymers, 2022, doi:10.3390/polym14245538_

Round 1

Reviewer 1 Report

There is no line numbering, making the review step much more difficult.

"It is a priority area for the major developed countries in the world." - Please insert reference.

"Commercially, Lithium-ion batteries (LIBs) are the most widely used secondary battery that plays an important role in our daily life." - please insert reference.

"Ji et al used a very simple method to synthesize PAQS (unit molecular structure of PAQS in Figure 2a) and investigate its electrochemical K+ storage performance for KIBs for the first time [33]." - please correct this citation (and others that are similar).

Correct the references that are inserted inside the tables.

There is a lack of topics for comparison, with a critical analysis of the processing and commercial value of the different materials that were cited in sections 2-4.

There are references with wrong formatting.

The writing of English should be improved, it is not in accordance with what is expected of a scientific document

Author Response

Response to Reviewer 1

Comments and Suggestions for Authors

1.Comment: There is no line numbering, making the review step much more difficult.

Response: We thank you for this comment. We have added line numbering in the revised manuscript.

  1. Comment: "It is a priority area for the major developed countries in the world." - Please insert reference.

Response: We thank you for this comment. We have inserted reference into the revised manuscript.

  1. Comment: "Commercially, Lithium-ion batteries (LIBs) are the most widely used secondary battery that plays an important role in our daily life." - please insert reference.

Response: We thank you for this comment. We have inserted references into the revised manuscript.

  1. Comment: "Ji et al used a very simple method to synthesize PAQS (unit molecular structure of PAQS in Figure 2a) and investigate its electrochemical K+ storage performance for KIBs for the first time [33]." - please correct this citation (and others that are similar).

Response: We thank you for this comment. We have corrected these citations in the revised the manuscript.

  1. Comment: Correct the references that are inserted inside the tables.

Response: We thank you for this comment. We have corrected the references that are inserted inside the tables.

  1. Comment: There is a lack of topics for comparison, with a critical analysis of the processing and commercial value of the different materials that were cited in sections 2-4.

Response: We thank you for this comment. Sections 2-4 focuses on the research and application status of polymer in potassium ion batteries systems. We also added some content in the revised manuscript.

  1. Comment: There are references with wrong formatting.

Response: We thank you for this comment. We have revised the references with wrong formatting.

  1. Comment: The writing of English should be improved, it is not in accordance with what is expected of a scientific document.

Response: We thank you for this comment. We have improved the writing of English in the revised manuscript.

Reviewer 2 Report

An interesting study is presented where the state of the art in a relevant topic is analyzed. It is indicated that in recent years, the research and development of batteries, is oriented to the application of Lithium and in this sense the KIB in the field of energy have received great attention. Compared to LIBs, KIBs have obvious cost advantages and are considered effective alternatives to LIBs, they have good application prospects in the field of large-scale energy storage batteries. Which translates into a relevant topic.

Line 84-87: For more information in this regard, a greater analysis of the state of the art is relevant.

The summary presented is relevant, some points associated with the comparison between Li and K, could be expanded the analysis of the state of the art. Associated with the methods of synthesis of the materials, the properties obtained, among others, which would impact on the information and relevance of the review.

Title could carry the word review.

Round 2

Reviewer 1 Report

After corrections are made in the review process, I recommend the manuscript for publication in Polymers.